# The effect of self-organizing map architecture based on the value migration network centrality measures on stock return. Evidence from the US market

Dariusz Siudak *

Institute of Management, Lodz University of Technology, Lodz, Poland

* dariusz.siudak@p.lodz.pl

## Abstract

Complex financial systems are the subject of current research interest. The notion of complex network is used for understanding the value migration process. Based on the stock data of 498 companies listed in the S&P500, the value migration network has been constructed using the MST-Pathfinder filtering network approach. The analysis covered 471 companies included in the largest component of VMN. Three methods: (i) complex networks; (ii) artificial neural networks and (iii) MARS regression, are developed to determine the effect of network centrality measures and rate of return on shares. A network-based data mining analysis has revealed that the topological position in the value migration network has a pronounced impact on the stock's returns.

**Data Availability Statement:** All relevant data are within the paper and its Supporting Information files.

## 1. Introduction

The value migration (VM) is the result of investors' seeking for effective capital allocation, increasing the stock's returns commensurate to the level of risk [1]. The knowledge of how value migrates between companies is the basis of value-based management in the considered set of enterprises. Value is a measure of the economic efficiency of a company. Since VM is the result of an evaluation of the company value carried out on the financial market, the proper measurement of value migration is an objective measure of the effectiveness of management in terms of adequate decision-making leading to increases in company value. An analysis of the VM processes enables effective value-based management and, at the same time, leads to continual reference to other companies in the considered set of enterprises. The value migration analysis conducted at various levels of aggregation allows to evaluate the effectiveness of capital allocation by investors; that is, an allocation based on expected return and estimated risk.

The VM process can be regarded as an evolving complex financial system consisting of many feedbacks. In this work, I have analyzed the VM process on the stock market through the application of complex networks. According to previous researches, complex networks are a powerful and widely used approach in econophysics field [2–10] to study the relationships among financial assets [11], where the financial system can be represented in a natural way as

**Funding:** The author received no specific funding for this work.

**Competing interests:** The authors have declared that no competing interests exist.

assets are vertices and the connections between them are links. A network-based data mining approach enables to reveal hidden information and relationships between assets that potentially affect market functioning. The network patterns embedded in the value fluctuations shed some new light on the corporate's share in the value fluctuation in the set of companies under consideration, i.e., entities listed on the stock exchange, and provide a deeper understanding of the VM process.

The main utilization of the correlation-based networks approach is to convert the multidimensional relationship matrix of the financial market into its sparse depiction. The stock correlation network is a subset of financial network that provides a deeper understanding of stock return time series [12–20], better predicts stock market behavior [21–27] and plays a significant role in portfolio optimization [28–35], risk assessment [36–39], asset allocation [40, 41]. In other words, correlation-based networks are a useful approach in economic decision-making [42] and can be regarded as the methodological basis of portfolio theory leading to efficient risk management.

It is important to note that many studies have found that the degree distribution of the cross-correlation network follows a power-law model [26, 43–56], and displays the non-fractal property [26, 54]. Furthermore, the degree distribution in the stock network based on mutual information [57] and Engle-Granger cointegration test [58] are also fitted to a power-law model.

The value migration network (VMN) is a network built on the basis of negative correlations between returns of daily stock price and the relative changes in the firms' market capitalization. In the VMN, companies (stocks) are represented as nodes, and the value flows between stocks are modeled as edges. It has been demonstrated that in- and out-degree distribution in the VMN obeys a power-law [59]. This implies that a relatively few hub-like assets strongly influence the fluctuations of the rest stock prices and corporate values in the entire stock market. As pointed out by Siudak [59], the VMN shows a disassortative behavior. In this light, only a few highly connected assets synchronize flows of value on the stock market in two directions, the inflow and outflow of value. However, it has been revealed [40] that the topological properties of financial networks vary at diverse time scales.

Dependency structures as well as the topological position of assets in the financial filtered network influence the rate of return on shares. Some studies in this area have recently discussed which stocks are superior–central or peripheral. Previous studies provide mixed results regarding the direction of the impact of the company's centrality in the cross-correlation network on stock returns. Some studies provide evidence of a positive relationship [36, 60–64], pointing out that central stocks are preferred, while some—of a negative one [65], indicating that it is better to invest in peripheries. Further research [66], using the minimum spanning trees based on linear correlation and mutual information, pointed out differentiated results, where the dominance of the stock type–central or peripheral–in terms of the impact on stock return depends on the assumed analysis period. On the one hand, due to the observed power-law behavior of degree distribution in the stock correlation network, vertices with many edges to other nodes (hubs) are preferentially located in the center of the network. These highly correlated stocks can drive the movements of the peripheral stock prices and the rates of return of central stocks prevail over the peripheral stocks. On the other hand, stocks lying in the networks' peripheral area have a more diversified structure and are less susceptible to uncontrolled stock price movements when the market is volatile. This applies to assets represented as a set of stocks across the entire stock market.

Other studies have focused on the network structure and its impact on the performance of the selected portfolio, where the risk-return ratio plays a crucial role and makes the analysis more complex. It is worth pointing out that assets located in different parts of the financial

filtered networks, especially peripheries, reveal different quotation patterns on the stock market, which can be beneficial to constructing an effectively diversified portfolio. Taking the return-risk relation into consideration, it has been revealed that peripheral portfolios dominate over central portfolios [32, 41, 67, 68]. The results of peripheral assets are superior to those of central ones, because portfolios composed of peripheral stocks have a more diversified structure and achieve a lower level of risk [66, 69, 70]. In other words, a more diversified portfolio outperforms central portfolio under return-risk relation. Further study using a clustering approach has shown that portfolio optimization can be performed using either peripheral stocks or central stocks depending on the stock market condition in the selection and investment horizon [71]. It should be stressed that network stability is an important element in portfolio construction [72].

However, the aforementioned studies have only concerned the correlation-based network approach, where the cross-correlation of log-return of stock price has been investigated. In this study, I consider the value migration network that is partly based on the Pearson correlation coefficients among assets, where only negative correlation coefficients are considered (see details in Section 2). A key aim of this work is to study the relationship between the topological position in the VMN and the annualized rate of return on shares in the S&P 500. Specifically, I examine the following research question: Does the company's topological position in the VMN determine the effect on individual stock return? The search for the economic factors influencing stock returns in the highly volatile conditions of the financial market is an open and up-to-date research area.

To the best of my knowledge, no previous work has endeavored to employ three powerful data mining methods in one study: 1) complex network, 2) artificial neural networks (ANN), and 3) multivariate adaptive regression splines (MARS). This work shows that the topological position in the value migration network, measured using a self-organizing features map, plays a crucial role in determining the stock's return. Based on the empirical analysis, it has been found that the company's topological position within the VMN has an impact on the annualized rate of return on shares. Specifically, centrality measures derived from the directed network, expressing the value flow pattern, are an appropriate indicator of annualized stock return. However, the direction of the impact of individual centrality measures is not specified; instead, a self-organizing map (SOM) of value migration based on the central position of the enterprise in the VMN has been created.

The main contribution of this work is twofold. First, I have produced a network illustrating the VM process. The network approach is important due to the possibility of extensive application of complex networks in the research process. The network dimension is part of a wide range of financial market research, extending the analysis to include the element of fluctuating company value.

Based on the measures of centrality of the value migration network, the SOM was obtained using the Kohonen network. Algorithm used in the unsupervised competitive learning process allows to detect hidden patterns of the VMN structure. This goal is accomplished without specifying the output signal defining the expected network reaction. The use of artificial neural networks enables the detection of the hidden topology of the value migration network.

Secondly, I employ MARS regression models to examine the impact of a self-organizing map in terms of the centrality position in the VMN on the annualized stock's returns. This network-based data mining analysis has the benefit of extending the framework of value-based management theory. My findings shed new insights on the VM process observed on the stock market and its impact on stock return. A better understanding of the value fluctuation patterns among assets in the network approach can be applied to network-based portfolio selection and risk management.

The remaining part of this paper is structured as follows. Section 2 explains the applied methodology. Section 3 briefly presents the empirical data. Section 4 includes the empirical results, and finally Section 5 provides a few conclusions.

## 2. Materials and methods

In this section, the research methodology is presented. The specified aim of the study has been performed employing a three-stage hybrid modeling procedure, an integration complex network and artificial neutral networks approaches with the multivariate adaptive regression splines technique. The diagram of the research process is shown in Fig 1.

First, I constructed the VMN based on i) negative Pearson correlation coefficients, used for identification connections between assets, and ii) the relative changes in market capitalization of each stock utilized to form the final direction of the relationship in the directed graph. The appropriate network construction procedure is presented in Section 2.1. The created value migration network is the basis for generating variables to search for the consequences of the network position of the enterprise and the network structure. Centrality measures (in- and out-degree; entropy; HITS centrality–see details in Section 2.5), taking into account the multifaceted position of the company in the network, were adopted as network variables.

Then I employed these centrality measures as the input variables in the input layer of SOM to detect the VMN topology based on the Kohonen network (Section 2.2). This topology was used to classify companies into disjoint groups taking into consideration specified neurons, using the entire data set (training, test, and validation set). The output layer of the Kohonen network consists of clusters with assigned companies due to the similarity according to the centrality measures. Then I sorted the groups (neurons) in a non-decreasing order according

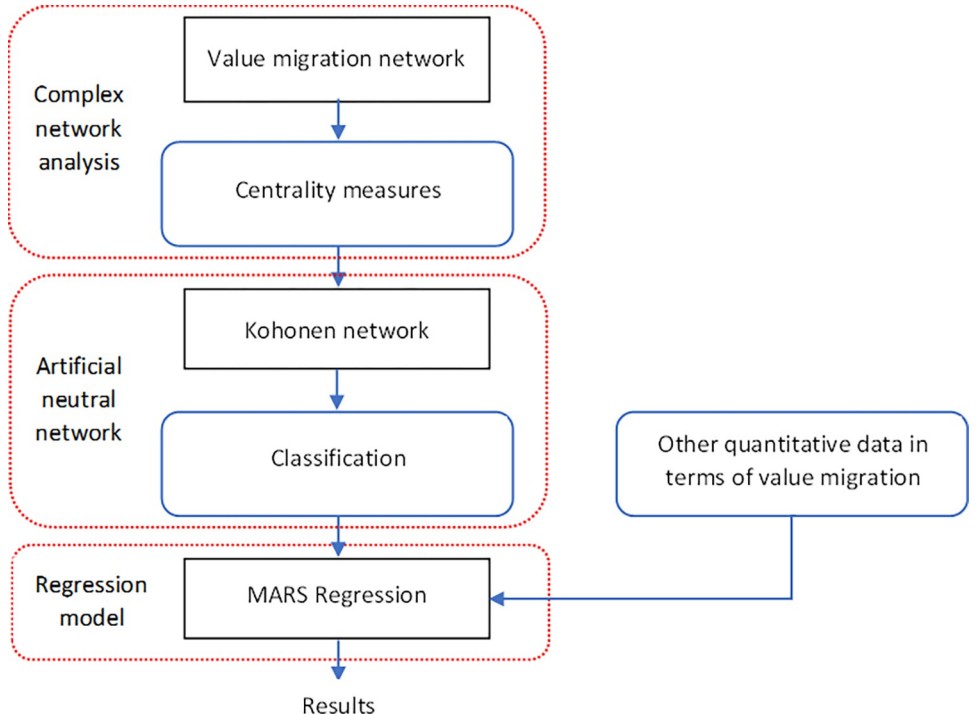

**Fig 1. General framework of the research process.**

to the average annualized log-rate of return on shares (ARRS) and assigned ranks also in a non-decreasing manner. As a result, an independent variable (clusters of centrality) is created. Finally, MARS regression was estimated to analyze relationships between the firm's structural position in the value migration network and the rate of return on shares (see details in Section 2.3). The dependent variable and independent variables used in the MARS regression are discussed in Sections 2.4 and 2.5, respectively.

## 2.1. Value migration network construction

The main issue is the construction of the VMN. Mantegna [12] proposed the minimal spanning tree (MST) method, the shortest tree linking all vertices in a graph, that has been a common and functional technique to filter out the information noise from the financial network, and has been widely utilized to examine stock return time series [73–81]. The advantage of MST is the simplification of network complexity [82] and effective compression of all information and investor expectations [83] by pruning the network consisting of $N(N-1)/2$ links to find the $N-1$ most relevant edges that connect all $N$ vertices without loops or cycles. The MST permits the tree structure, whereas other filtering methods like the Planar Maximally Filtered Graph (PMFG), asset graph, or the winner-takes-all allow to form loops. For this reason, the MST approach is the most appropriate for visualizing the VM process as a graph.

The robustness problem regarding the non-uniqueness of the MST should be emphasized when the system contains more than one MSTs [84]. The MST is unique when all link weights between each pair of nodes are dissimilar [85], which can only be fulfilled when $N$ is large and $T \gg N$ [86], where $N$ is the number of nodes and $T$ is the length of time series. To preserve the uniqueness of the minimum spanning tree, the MST-based Pathfinder approach [87] has been applied to construct the VMN. This PFNET algorithm parametrized with ($r = \infty$ and $q = N-1$) directly prunes a given network as the union of all MSTs possibly existing in the original network is extracted. In other words, Pathfinder directly merges all edges with reference to the separately existing MSTs of the original network that is equivalent to the PFNET ($\infty$; $N-1$). The following steps based on the procedure presented in Ref. [59] are performed to build the value migration network:

1. Individual time series related to the company are selected.

2. Based on the closing stock-price, the $N$ x $N$ symmetric matrix of the Pearson correlation coefficient **C** between log-returns of pairs of stocks is determined:

$$c_{ij} = \frac{\langle r_i r_j \rangle - \langle r_i \rangle \langle r_j \rangle}{\sqrt{(\langle r_i^2 \rangle - \langle r_i \rangle^2)(\langle r_j^2 \rangle - \langle r_j \rangle^2)}} \tag{1}$$

   where

$$\langle r_i \rangle = \frac{1}{T} \sum_{t=1}^{T} r_i(t) \tag{2}$$

$$r_i = \ln P_i(t) - \ln P_i(t-1) \tag{3}$$

   for $i,j = 1, 2, \ldots, N$ and $t = 1, 2, \ldots, T$.

3. The cross-correlation matrix **C** is pruned to the $N$ x $N$ matrix form

$$\mathbf{C} = [c_{ij}'] \tag{4}$$

where the only negative correlation coefficients is considered as the following equation:

$$c'_{ij} = \begin{cases} c_{ij} \ if \ c_{ij} < 0 \\ 0 \ otherwise \end{cases} \tag{5}$$

4. The yearly logarithmic return of the firm's market capitalization is calculated:

$$R_{MV_i}(t) = \ln MV_i(t) - \ln MV_i(t-1); \ for \ i = 1, 2, \ldots, N \tag{6}$$

where $MV_i(t)$–market capitalization of asset $i$ at time $t$; $t$–end of the year; and the return of market capitalization for entire market is computed:

$$R_{\sum MV_i(t)} = \ln \sum_{i=1}^{N} MV_i(t) - \ln \sum_{i=1}^{N} MV_i(t-1) \tag{7}$$

The annual window is utilized to diminish the effects of short-term fluctuations in the company value.

5. The specific threshold value $\theta = 0.6667$ is set.

6. Each stock is assigned to one of the two possible phases, (1) inflow or (2) outflow, by means of the following formulas: inflow phase:

$$if \ R_{MV_i}(t) \geq \theta R_{\sum MV_i(t)} \tag{8}$$

and outflow phase:

$$if \ R_{MV_i}(t) < \theta R_{\sum MV_i(t)} \tag{9}$$

7. Based on the reduced cross-correlation matrix **C'**, the polytomous variable $x_{ij}$ is defined as follows:

$$x_{ij} = \begin{cases} 1 \ if \ i \in Inflow \ phase \ and \ j \in Outflow \ phase \\ -1 \ if \ i \in Outflow \ phase \ and \ j \in Inflow \ phase \\ 0 \ if \ otherwise \end{cases} \tag{10}$$

which can be formulated in an equivalent way:

$$x_{ij} = \begin{cases} 1 \ if \ R_{MV_i}(t) \geq \theta R_{\sum MV_i(t)} \ and \ R_{MV_j}(t) < \theta R_{\sum MV_j(t)} \\ -1 \ if \ R_{MV_i}(t) < \theta R_{\sum MV_i(t)} \ and \ R_{MV_j}(t) \geq \theta R_{\sum MV_j(t)} \\ 0 \ if \ otherwise \end{cases} \tag{11}$$

where

$$R_{\sum MV_i(t)} = R_{\sum MV_j(t)} \tag{12}$$

8. The symmetric matrix **A** with dimensions $N$ x $N$ is constructed based on the elements $c_{ij}'$ of matrix **C'** and variable $x_{ij}$ as follows:

$$A = [a_{ij}] = |c_{ij}' \cdot x_{ij}| \text{ for } i, j, = 1, 2, \ldots, N, \tag{13}$$

where $a_{ij}$ is in the range [0; 1], formulating an undirected graph.

9. Based on the matrix **A**, a simple nonlinear transformation

$$d_{ij} = \sqrt{2(1 - a_{ij})} \text{ for } i, j, = 1, 2, \ldots, N \tag{14}$$

within the range [0; $\sqrt{2}$] is executed using the metric distance that satisfies the axioms of a metric distance–a) nonnegativity; b) symmetry; c) triangular inequality, forming the $N$ x $N$ symmetric distance matrix **D** = [$d_{ij}$].

10. Appling the MST-Pathfinder approach [87] in conjunction with the Kruskal's algorithm [88], the matrix **D** is pruned where the sum of all edge distances ($d_{ij}$) in the tree is minimized. As a result, the PFNET ($\infty$, $N$-1) network is extracted as the unification of all MSTs that possibly exist in the matrix **D**, denoted by **T** = [$t_{ij}$].

11. The PFNET($\infty$, $N$-1) network is then transformed to a dichotomized matrix **T'**, that elements $t_{ij}'$ are defined as

$$t_{ij}' = 1 \ if \ t_{ij} > 0 \tag{15}$$

12. In the final step, the direction of value migration is specified. An adjacency matrix **A'** is obtained by giving the direction of the network whose elements are computed using the following transform rule:

$$A = [a_{ij}'] = \begin{cases} \left.\begin{array}{l} a_{ij}' = 1 \\ a_{ji}' = 0 \end{array}\right\} & if \ t_{ij}' = t_{ji}' = 1 \ and \ x_{ij} = -1 \\ \left.\begin{array}{l} a_{ij}' = 0 \\ a_{ji}' = 1 \end{array}\right\} & if \ t_{ij}' = t_{ji}' = 1 \ and \ x_{ij} = 1 \end{cases} \tag{16}$$

where $x_{ij}$ = −1 means that $i \in$ *Outflow phase* and $j \in$ *Inflow phase*
$x_{ij}$ = 1 means that $i \in$ *Inflow phase* and $j \in$ *Outflow phase*.

The presented procedure allows us to generate the VMN that is unweighted, directed, planar, without a cycle or self-edges, simple graph. It should be emphasized that creating a connection between a pair of stocks in the VMN is possible under two conditions: i) the occurrence of a negative Pearson correlation coefficient among pair of stocks; ii) both assets have to be assigned to opposite phases of value migration. It is not possible to form a connection between two companies if both firms are in the same value migration phase or if there is a positive correlation coefficient between them.

## 2.2. Kohonen network

Kohonen network is a type of artificial neutral network first proposed by Kohonen [89, 90], that has the ability to create the self-organizing map according to environmental features. Kohonen network is an unsupervised learning, self-organizing, competitive network. The self-organizing map is an automatic data-analysis approach, widely used in clustering problems

and data exploration [91]. The aim of the Kohonen network is to detect similarities among data and cluster data set into different classes using a non-linear, unsupervised, and competitive learning algorithm. In other words, similar objects in the data structure are automatically clustered and nonlinear mapping of individual classes follows a learning pattern recognized by specified neurons without determining target variables. Kohonen network attempts to learn the data structure by using the features contained in the data set in a hidden way, including the input variables without network reaction as a predefined output signal. It is used to search for the division of a given set of objects into homogeneous subsets without knowing the pattern of classes. The result of the Kohonen network's learning process is a topological map that allows to determine the importance of the relevant regions of the map formed by neurons in the topological layer of the network.

The SOM does not contain a hidden layer and consists of an input layer and an output layer, the last of which is called competitive layer or topological layer. The input layer is unidimensional and contains the number of neurons corresponding to the dimensions of the input eigenvector. The competitive layer is bidimensional, and the nodes in the output layer are the equivalent of output neurons, which are distributed in a regular two-dimensional grid, curved in the $N$-dimensional input space. These two layers–input and output–are linked in two directions where all input nodes are connected to all output nodes using the weights of the synapses. These weights can be expressed as $w_{ij}(i = 1,2,\ldots,n; j = 1,2,\ldots,m)$, where $n$ and $m$ are the number of nodes in input layer and competition layers, respectively. The more similar the input objects are, the closer to each other the corresponding neurons will be located in the space of the topological layer.

The iterative Kohonen network learning algorithm can be summarized as follows [91–93]:

1. Parametrization. Setting the parameters preceding the execution of the iterative procedure: i) a topological length and width of the network defining the discrete output space; ii) a learning rate parameter; iii) a neighborhood radius parameter; iv) number of epochs.

2. Initialization. Selection of the connecting weights from the available set of input nodes in a random manner.

3. Data input. The data of sample is entered into the input layer after normalization

$$x^k = (x_1, x_1, \ldots, x_n) \tag{17}$$

4. Distance computation. The Euclidean distance of connection weight between input nodes and output neurons is computed:

$$s_j = \sum_{i=1}^{n} (x_i^k - w_{ij})^2, (j = 1, 2, \ldots, m) \tag{18}$$

where $x_i$–input value of $i$th input node.

5. Similarity matching. Finding the best-matching neuron (winning node) with the minimum Euclidean distance:

$$c = \arg \min_{j \in (1,2,\ldots,m)} \{s_j\} \tag{19}$$

6. Weight update. Adjustment of the synaptic weigh vectors of all neurons applying to the adjustment formula:

$$w_{ij}(t + 1) = w_{ij}(t) + h_{c,j}(t) \cdot [x_i^k(t) - w_{ij}(t)] \tag{20}$$

where: $h_{c,j}(t)$−neighborhood function centered around the winning node $c$:

$$h_{c,j}(t) = \eta(t) \cdot \exp\left(\frac{-r_{c,j}^2}{2\delta(t)^2}\right) \tag{21}$$

where $r_{c,j}$−the geometric distance between winning neuron and $j$th node; $\delta(t)$−neighborhood radius monotonically decreasing with the learning process; $\eta(t)$−learning rate in the range (0; 1):

$$\eta(t) = \eta_{min} - \frac{t}{T} \cdot (\eta_{max} - \eta_{min}) \tag{22}$$

where $t$−current number of iterations; $T$−number of epochs.

7. Continued training. Input of the data of the next sample and a continued stepwise recursive procedure with step 2, until all samples in the training set have been trained and all epochs have been executed.

The research sample was divided into three sets of data in a proportion of 70% for the training set, 15% for the test set, and 15% for the validation set. The training set is employed to construct the Kohonen network, the test set−to evaluate the quality of the classification as the ability to generalize, while the validation set−to verify the applicability of the created SOM model in the network training process. The SOM quality evaluation is performed using the quantization error, which measures the mean distance between each data case and its corresponding winning neuron

$$Q_{error} = \frac{1}{N} \sum_{i=1}^{n} (x_i^k - w_{c,i})^2 \tag{23}$$

where: $w_{c,i}$−weight of the winning neuron corresponding to the input value of the $i$th input node.

The quantization error assesses the fit of the topology map to the data. The best result of the Kohonen network algorithm is yielded for the minimum quantization error value.

In order to determine the SOM, the following parameters were adopted: i) learning rate: Start = 0,1, End = 0,02; ii) neighborhood radius: Start = 3; End = 0; iii) number of Epochs: 200; iv) network initialization: random, Gaussian; v) topological network length and width: a combination of length/width from 3 to 6.

As a result, 10 networks were created to test the dimensions of the topological map from 3 x 3 to 6 x 6. The projection of the self-organizing feature map is the basis for the further analysis process using the MARS regression approach.

## 2.3. Multivariate adaptive regression splines method

Multivariate adaptive regression splines is a non-parametric and nonlinear regression method, first developed by Friedman [94]. It is an adaptive procedure for effective modeling of complex relationships embedded in multivariate datasets. Furthermore, MARS has the ability to reliably track complex data structures across all degrees of interactions. A regression model is automatically created by considering the nonlinear interaction between variables by fitting the data into a series of spline functions. MARS regression has several advantages with respect to the ability to model complex relationships among variables, namely: (1) no prior assumptions are required for the underlying relationships between a dependent variable and a set of independent variables, as well as the distribution of these variables; (2) quantitative and qualitative explanatory variables can be included in the model; (3) the ability to flexibly model the

nonlinearity effects and predictors interactions; (4) any degree of interactions between explanatory variables is allowed; (5) unlike other regression techniques, there is no dimensionality problem.

MARS model is a nonlinear relationship between the dependent variable and the explanatory variables by means of applying a set of piece-wise regression spline functions implemented by a sequence of linear polynomial basis functions in distinct intervals of the independent variable space. The spline basis functions use two-sided truncated reflected pairs [95]:

$$(x - t)_+ = \begin{cases} x - t, & if \ x > t \\ 0, & if \ x \le t \end{cases} \tag{24}$$

$$(t - x)_+ = \begin{cases} t - x, & if \ x < t \\ 0, & if \ x \ge t \end{cases} \tag{25}$$

where: $t$ is the knot point of basis function, index "+" means positive part of the basis function.

The slope of the regression line is expressed over an interval and changes when the two knot points is crossed. This means that the explanatory variable may enter the model with a different sign and a different value of the coefficient depending on which side of the knot point is its value.

The MARS model for dependent variable $y$ with $M$ spline functions can be described as [95]:

$$y = f(x) = \beta_0 + \sum_{m=1}^{M} \beta_m h_m(x) \tag{26}$$

where: $\beta_0$–constant; $\beta_m$–estimated coefficients of basis functions; $M$–number of spline basis functions; $h_m(x)$–single spline function or a product of two or more basis functions for distinct predictors ($x$) [94]:

$$h_m(x) = \prod_{k=1}^{k_m} [s_{km}(x_{v(k,m)} - t_{k,m})]_+ \tag{27}$$

$k_m$–number of knot points; $s_{km}$–+1 (-1) values that indicate right (left) sense of the associated step function; $v(k,m)$–label of the predictors ($x$); $t_{k,m}$–knot location.

The fitting of the coefficients of spline basis functions to the data is performed using the standard OLS procedure [94]. The maximum possible number of all spline basis functions is $2Np$, where $N$ is the number of observations corresponding to the number of vertices in the VMN network, and $p$ is the number of explanatory variables.

Then, an intensive search procedure is carried out for selecting explanatory variables and estimating the knot points for each variable. The two-stage process is performed to optimize the final form of MARS model. In the first stage, forward selection is carried out by successively adding reflected pairs of basis functions to the model. As a result, a large number of basis functions is included into the model, leading to overfitting the data. Then, the backward deletion phase procedure is initiated. The over-fit model is pruned by removing the spline functions in order of least contribution to the accuracy of the model fit based on the generalized cross-validation (GCV) criterion [96] defined as a quotient of the mean squared residual error and a penalty of the model complexity [97]:

$$GCV(M) = \frac{\frac{1}{N} \sum_{i=1}^{N} (y_i - \hat{y})^2}{\left[1 - \frac{C(\check{M})}{N}\right]^2} \tag{28}$$

where: $C(\tilde{M}) = C(M) + d \cdot M$; $d$–penalty factor; $C(M)$–number of parameters being fit; $y_i$– observed dependent variable; $\hat{y}$—predicted target variable.

During the backward procedure, the least significant basis functions are removed from the MARS model in terms of the measure of fit. The model reduction in the backward sequence is stopped when the minimum of the $GCV(M)$ is attained.

Parameters for the MARS regression analysis were individually set depending on a specific model. These parameters are presented in the Section 4, where the results are presented. The variables were normalized prior to the execution of the MARS models using the following transformation function:

$$x_{ij}^N = \frac{x_{ij}}{\max_i\{x_{ij}\} - \min_i\{x_{ij}\}} \left( \max_i\{x_{ij}\} - \min_i\{x_{ij}\} \neq 0 \right) \tag{29}$$

The normalized variable is in the range

$$x_{ij}^N = \left\langle \frac{\min_i\{x_{ij}\}}{\max_i\{x_{ij}\} - \min_i\{x_{ij}\}}, \frac{\max_i\{x_{ij}\}}{\max_i\{x_{ij}\} - \min_i\{x_{ij}\}} \right\rangle \tag{30}$$

and this range is constant, equal to one.

## 2.4. Dependent variable

The dependent variable is the annualized log-rate of return on shares

$$ARRS_i = \ln P_i(t) - \ln P_i(t - \Delta t) \tag{31}$$

for all $i = 1, 2, \ldots, N$, where $ARRS_i$–annualized rate of return on shares $i$; $\Delta t$–one year.

## 2.5. Independent variables

Three measures were used as dependent variables in the MARS regression analysis. Two of them relate to measures of VM process, and one variable is aggregated based on the centrality measures of VMN.

### 2.5.1. Value migration measures.

(i) Share in value migration balance ($SVMB_i$):

$$SVMB_i = \frac{\Delta MV_i}{|\sum_{i=1}^N \Delta MV_i|} \left( \sum_{i=1}^N \Delta MV_i \neq 0 \right) \tag{32}$$

where $\sum_{i=1}^N \Delta MV_i$ is the balance of VM defined as the sum of the inflows and the outflows of the company's market capitalization; $MV_i$–market capitalization of the companies $i = 1, 2, \ldots, N$.

The flows in a company's value may compensate for one another, which in result leads to a zero balance of value migration of the considered set of corporates within one period of analysis. Hence, one needs to introduce a limiting condition:

$$\sum_{i=1}^N \Delta MV_i \neq 0 \tag{33}$$

If $\sum_{i=1}^N \Delta MV_i = 0$, then the share in migration balance, calculated separately for each company, will be undetermined. The variable share in the value migration balance ($SIVMB_i$) was adapted and reformulated from the work [98].

(ii) $MV_{i,(t-1)}$–the market capitalization of the company $i$ at the beginning of the measurement period of the VM processes.

### 2.5.2. Network measure.

(iii) Cluster of centrality ($CC_i$)–is an independent variable identifying the group to which the company has been qualified by the Kohonen neutral network based on centrality measures of the value migration network. This variable is the result of applying two research methods: complex network and Kohonen neutral network.

**2.5.3. Centrality measures.** Because network centrality has a significant role in determining the internal structure of a financial network [99], in order to assign a company to the appropriate neuron (variable: cluster of centrality), I used four centrality measures dedicated to the directed network. The applied centrality measures reflect dissimilar premises and take into account a different type of central position of enterprise in the VMN. Pozzi, Di Matteo and Aste [32] pointed out that the different sensitivity of individual centrality measures to outliers and noise justifies the combined use of these centrality indices instead of utilizing each of them in isolation. The following four centrality measures are the input variables in the input layer of the Kohonen network:

i.  In-degree centrality–the portion of vertices adjacent to the node $i$

$$DC_i^{in} = \frac{k_i^{in}}{N-1} \tag{34}$$

ii.  Out-degree centrality–the portion of vertices adjacent from the vertex $i$

$$DC_i^{out} = \frac{k_i^{out}}{N-1} \tag{35}$$

iii.  Entropy centrality–which is related to the distribution of the probabilities that the flow stops at each of the vertices in the graph [100], defined as

$$C_i^H = \frac{-\sum_{\substack{j \in V \\ p_{ij} \neq 0}} p_{ij} \log p_{ij}}{\log N} \tag{36}$$

where

$$p_{ij} = -\sum_{k=1}^{K(i,j)} \sigma_k(j) \prod_{t=0}^{n(k)-1} \tau_k(v_t); \ \sigma_k(v_t) \tag{37}$$

is stopping probability and $\tau_k(v_t)$ denotes transfer probability.

iv.  HITS centrality. The HITS approach–hyperlink-induced topic search algorithm developed by Kleinberg [101], where each vertex in a network has two types of centrality: an authority centrality and a hub centrality. HITS centrality expresses the difference between an authority centrality and a hub centrality

$$HITS_i = ac_i - h_i \tag{38}$$

where:

$$ac_i = \alpha \sum_j a'_{ij} \cdot h_j \tag{39}$$

$$h_i = \beta \sum_j a'_{ij} \cdot ac_j \qquad (40)$$

$a'_{ij}$–elements of the adjacency matrix $\mathbf{A}'$; $ac_i$ ($ac_j$)–authority centrality of vertex $i$ ($j$); $h_i$ ($h_j$)–hub centrality of vertex $i$ ($j$); $\alpha, \beta$ –constant.

Authority centrality and hub centrality are in the interval $\langle 0; 1 \rangle$, which means that the HITS centrality is in the range $\langle -1; 1 \rangle$. In the VMN value can flow in or out of any company. If the value flows into the enterprise, then vertex has only in-neighbor, non-zero authority score, hub centrality equals zero, and HITS centrality has a positive value of (between 0 and 1). If the value flows out of the company, then vertex has only out-neighbor, non-zero hub centrality score, authority centrality is zero, and HITS centrality has a negative value of (between -1 and 0).

## 3. Data set

I used a dataset consisting of daily closure prices of 498 stocks that were continuously listed in the S&P 500 index between the end of 2018 to the end of 2019 (253 daily closing prices in each time-series). The data were obtained from Yahoo Finance [102]. In order to construct the value migration network, data on the market capitalization of companies at the end of 2018 and 2019 were used. This means that in the design of the VMN, the length of the time-window of one year was used to explore the fluctuating market value of the company. The data were retrieved from Refinite database [103]. The data used in further analyses was included in the supplementary material associated with this article (S1 Dataset).

The value migration network is shown in Fig 2.The VMN consists of 498 vertices and is a directed and unweighted network. The number of connected nodes within the largest component is 471 and the number of edges is $N$-1 = 470, which is characteristic of MST-based networks. The remaining 27 companies are isolated nodes, which means that their source (sink) of the inflow (outflow) of value is placed outside the analyzed set of companies. The analysis covered 471 companies included in the largest component of VMN.

All network analyses were carried out in NetMiner software [105] while Kohonen network and MARS regression were performed in Statistica software [106].

Several previous studies based on data from the US stock market have shown that one company–General Electric (GE)–is the hub vertex and occupies the central position in the correlation-based MST [12, 15, 66, 74, 75, 107, 108] and PMFG [107]. The above results have been confirmed in another study [109], in which two MST-based networks were constructed for the time series of stock returns in 1998–1999 for the 1000 and 100 largest US companies, respectively.

In this paper, the American Water Works Company (AWK) is the center of the value migration network, which has the second largest number of sources of value inflow, after Newmont Corporation (NEM); $k_i^{in}$ is 33 and 36, respectively. General Electric is not found to be the hub, although it is in the inflow phase; $k_i^{in} = 2$. It should be noted that there are significant differences in the construction of the VMN and the cross-correlation network.

## 4. Results and discussion

This section reports the impact of centrality measures of the VMN on the annualized stock return. After determining the four centrality measures for the value migration network, a neutral Kohonen network was applied. The dimension of the SOM from 3 x 3 to 6 x 6 was tested and the most appropriate one was determined based on the minimum error level criterion for

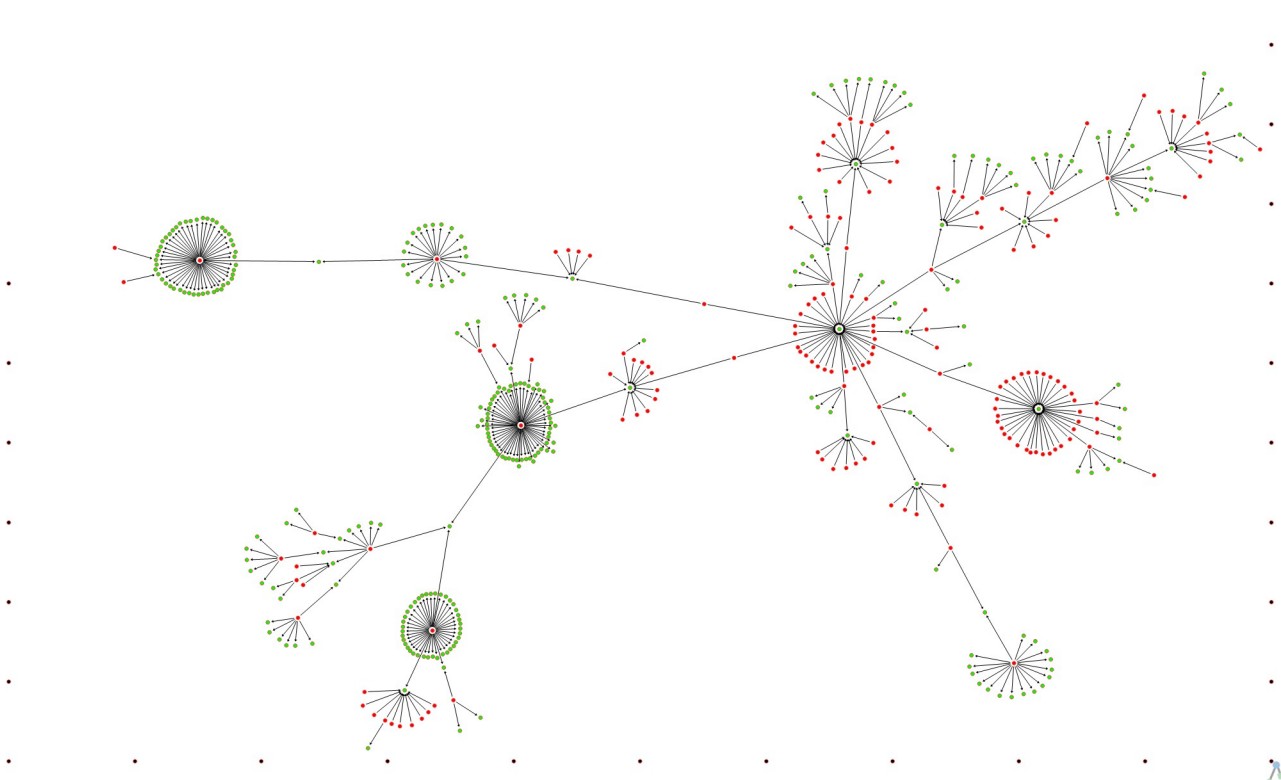

**Fig 2. Value migration network.** Green nodes denote companies in the inflow stage; red vertices indicate firms in the outflow stage; black nodes are isolated (visualization algorithm: Fruchterman & Reingold [104]; performed in NetMiner [105]).

the test and validation sets. The error level for the training set is less important because ANN tends to over-fit the model to the data (network overfitting) in the network learning process, hence the mean error may be close to zero. In the network assessment, the most important information is the value of the validation error obtained from the validation set. An additional test of the network model is performed using a test set to ensure that the achieved error levels for the training and validation sets are correct regardless of the mechanism of the network training process.

Of the possible 10 networks presented in Table 1, the lowest validation error level is achieved for the 3 x 4 network dimension, and the value of test error is close to the value of error obtained from the training and validation set. Because of this, as well as a lack of over-fitting or the inability to generalize, SOM with a topological layer size of 3 x 4 is used for further analysis.

The numbers of firms classified into disjoint clusters in the self-organizing map for the separate data sets are presented in Table 2 and Fig 3A. Fig 3B shows the distance map between neurons.

For the combined training, test and validation sets, a total of 8 neurons in the topological layer were obtained: (1, 1); (2,1); (3,1); (3,2); (1,3); (1,4); (2,4),(3,4). This represents the topological map of the centrality level for the entire dataset based on the value migration network. Table 3 presents the number of firms included in separate clusters, as well as the proportion and mean annualized log-rate of return on shares (mean $ARRS_i$). The clusters were ordered in

**Table 1. Errors for subsequent networks.**

| Network Id | Network* | Topological layer size | Quantization error | | |
|---|---|---|---|---|---|
| | | | training | test | validation |
| 1 | SOM 4–9 | 3x3 | 0.00444 | 0.00318 | 0.00301 |
| **2** | **SOM 4–12** | **3x4** | **0.00308** | **0.00319** | **0.00225** |
| 3 | SOM 4–16 | 4x4 | 0.00173 | 0.00347 | 0.00311 |
| 4 | SOM 4–20 | 4x5 | 0.00132 | 0.00377 | 0.00359 |
| 5 | SOM 4–25 | 5x5 | 0.00078 | 0.00491 | 0.00385 |
| 6 | SOM 4–15 | 3x5 | 0.00217 | 0.00324 | 0.00297 |
| 7 | SOM 4–18 | 3x6 | 0.00141 | 0.00367 | 0.00335 |
| 8 | SOM 4–24 | 4x6 | 0.00113 | 0.00405 | 0.00384 |
| 9 | SOM 4–30 | 5x6 | 0.00044 | 0.00439 | 0.00309 |
| 10 | SOM 4–36 | 6x6 | 0.00060 | 0.00442 | 0.00333 |

*—the first number indicates the number of variables, the second number is the size of the output

The network selected for further analysis is highlighted in bold

a non-descending order with respect to mean $ARRS_i$ and ranks were assigned–see the last column in Table 3. The higher the mean $ARRS_i$ in the group, the higher the rank.

Then, two regression models, which took the network structure into consideration as an independent variable, were constructed for which the *cluster of centrality* ($CC_i$) was assumed. The dependent variable is the annualized rate of return on shares $i$ ($ARRS_i$). The explanatory variables are cluster of centrality ($CC_i$), share in value migration balance ($SVMB_i$), and the market capitalization of company $i$ at the beginning of the measurement period of value migration processes ($MV_{(t-1)}$).

The first model was constructed on the basis of three explanatory variables. The value of the maximum number of spline basis functions is restricted to 14, the penalty factor equals 2, and no interactions between the variables were allowed. Table 4 reports the parameters and the results obtained for model 1.

MARS model contains 7 basis functions that are a single spline specified by only one predictor. The value of the overall Generalized Cross Validation (GCV) statistic is 0.0068, and the adjusted coefficient of determination (*adjusted $R^2$*) is 0.5856. The value of GCV criterion close to zero indicates the reliability of the model. The explanation of the variance exceeds 58.5%. It can be seen from Table 4 that all regression coefficients of the contained spline basis functions are statistically significant at a significance level of 1% or less, which implies that all variables play crucial roles in determining the annualized stock's return.

In Table 4, the regression coefficients (β) are given for each basis function. The first two basis functions BF1 and BF2 relate to the non-linear effect of the cluster of centrality in the VMN. The value of the knot point of BF1 and BF2 is 5 (the original value of 0.714 normalized knot). This means that the critical threshold of the group to which the company has been

**Table 2. Representation of companies in SOM 3 x 4 for specific datasets.**

| Sample: | Training | | | | Test | | | | Validation | | | | Training, Test, Validation | | | |
|---|---|---|---|---|---|---|---|---|---|---|---|---|---|---|---|---|
| Cluster position | 1 | 2 | 3 | 4 | 1 | 2 | 3 | 4 | 1 | 2 | 3 | 4 | 1 | 2 | 3 | 4 |
| 1 | 2 | 0 | 94 | 16 | 0 | 0 | 15 | 4 | 0 | 0 | 20 | 5 | **2** | **0** | **129** | **25** |
| 2 | 39 | 0 | 0 | 14 | 14 | 0 | 0 | 1 | 8 | 0 | 0 | 3 | **61** | **0** | **0** | **18** |
| 3 | 6 | 159 | 0 | 1 | 0 | 35 | 0 | 1 | 3 | 30 | 0 | 1 | **9** | **224** | **0** | **3** |

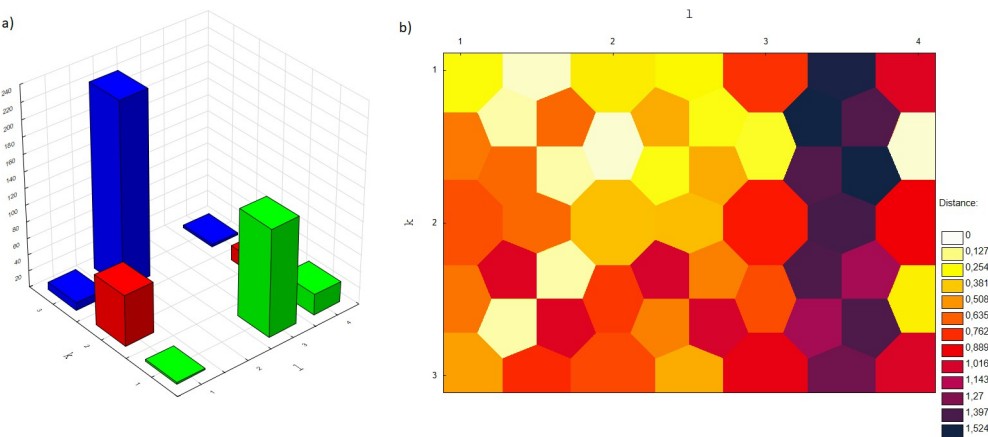

**Fig 3.** The numbers of firms classified into disjoint clusters (a) and the distances between neurons (b) for SOM 3 x 4.

qualified by the Kohonen neutral network based on centrality measures of the value migration network occurs at number 5. The effect of the cluster of VMN can be diverse: (1) if the company is classified to group number 6, 7, or 8, then BF1 indicates a positive effect on the stock's return; (2) if the company is classified to group number from 1 to 4, then BF2 indicates a negative effect on the stock's return. For companies included in group 5, centrality in the VNM network does not affect the generated rate of return on shares. It turns out that it is a neuron (1, 1) containing only 2 enterprises (see Table 3).

However, taking into account three elements: (i) the form of spline basis functions–BF1: ($x$-$t$); BF2: ($t$-$x$)–(ii) the sign (+x or -x) standing next to the dependent variable and (iii) positive regression coefficient for BF1 or negative for BF2, the direct direction of changes in $CC_i$ and $ARRS_i$ is the same, as can be summarized using appropriate notation

for BF1:

$$\text{for } CC_i > 5 \text{ and } \nearrow CC_i \longrightarrow \nearrow BF1 \longrightarrow \nearrow ARRS_i$$

$$\text{for } CC_i > 5 \text{ and } \searrow CC_i \longrightarrow \searrow BF1 \longrightarrow \searrow ARRS_i$$

for BF2:

$$\text{for } CC_i < 5 \text{ and } \nearrow CC_i \longrightarrow \searrow BF1 \longrightarrow \nearrow ARRS_i$$

$$\text{for } CC_i < 5 \text{ and } \searrow CC_i \longrightarrow \nearrow BF1 \longrightarrow \searrow ARRS_i$$

where $\nearrow$ and $\searrow$ denote the increase and decrease of the variable, respectively.

**Table 3. Average value of ARRS for individual neurons and their ordering.**

| Neuron | n | Proportion | Cumulative Proportion | Mean $ARRS_i$ | Cluster of centrality ($CC_i$) |
|---|---|---|---|---|---|
| (2,4) | 18 | 0.038 | 0.038 | -0.0342 | 1 |
| (1,4) | 25 | 0.053 | 0.091 | -0.0055 | 2 |
| (3,4) | 3 | 0.006 | 0.097 | 0.0525 | 3 |
| (1,3) | 129 | 0.274 | 0.371 | 0.0572 | 4 |
| (1,1) | 2 | 0.004 | 0.375 | 0.2645 | 5 |
| (3,2) | 224 | 0.476 | 0.851 | 0.3149 | 6 |
| (3,1) | 9 | 0.019 | 0.87 | 0.3153 | 7 |
| (2,1) | 61 | 0.13 | 1 | 0.3772 | 8 |

**Table 4. Parameters and results for model 1.**

| MODEL 1 | | | | |
|---|---|---|---|---|
| **Parameters** | | Value | | |
| **Dependent variable** | | $ARRS_i$ | | |
| **Independent variables** | | $CC_i$; $SVMB_i$; $MV_{(t-1)}$ | | |
| **Maximum number of basis functions** | | 14 | | |
| **Order of interaction** | | 1 | | |
| **Penalty** | | 2 | | |
| **Threshold** | | 0.0005 | | |
| **Removal of irrelevant basis functions** | | Yes | | |
| **Number of observations** | | 471 | | |
| **Basis function** | | $\beta$ | $t$ statistics | $p$ |
| | Constant | 0.135 (0.011) | 12.145*** | 0.0000 |
| **BF1** | $max(0, CC_i - 0.714)$ | 0.238 (0.034) | 6.927*** | 0.0000 |
| **BF2** | $max(0, 0.714 - CC_i)$ | -0.255 (0.032) | -7.881*** | 0.0000 |
| **BF3** | $max(0, SVMB_i - 0.014)$ | 2.747 (0.727) | 3.777*** | 0.0002 |
| **BF4** | $max(0, 0.014 - SVMB_i)$ | -4.618 (0.492) | -9.392*** | 0.0000 |
| **BF5** | $max(0, MV_{(t-1)} - 0.049)$ | -0.209 (0.060) | -3.461*** | 0.0006 |
| **BF6** | $max(0, 0.049 - MV_{(t-1)})$ | 1.228 (0.306) | 4.018*** | 0.0001 |
| **BF7** | $max(0, SVMB_i - 0.035)$ | -2.392 (0.739) | -3.237** | 0.0013 |
| **Results** | | Value | | |
| **GCV** | | 0.0068 | | |
| $R^2$ | | 0.5927 | | |
| **Adjusted $R^2$** | | 0.5856 | | |

standard error in parentheses

***, ** and * denote significant levels at 0.1%, 1%, and 5%, respectively

This means that regardless of the initial value of $CC_i$, companies from the cluster with a higher index ($CC_i$) have a higher rate of return on shares. In other words, if $CC_i$ is greater than 5, then the greater the value of $CC_i$, the greater the $ARRS_i$, and vice versa. If $CC_i$ is less than 5, the greater the $CC_i$ value (up to the limit of $CC_i = 4$), the smaller the decrease in $ARRS_i$, and if the smaller the $CC_i$ value, the greater the decrease in $ARRS_i$.

The basis functions BF3 and BF4 refer to the share in the value migration balance, where the critical threshold is 0.0014 (the original value of normalized knot point is 0.0141). If the share in the VM balance is greater than 0.14%, then the impact on stock return is positive and vice versa. On the other hand, BF7 indicates that if the value of $SVMB_i$ is greater than 0.347% (the original value of 0.0348 normalized knot point), then the share in the VM balance has a negative effect on the annualized stock return. Comparing the similar forms of the spline basis functions BF3 and BF7 ($x$-$t$), it can be seen that they have opposite signs for the regression coefficient–positive and negative, respectively. From the theoretical point of view, a larger share in the VM balance should contribute to achieving a higher rate of return on shares. Hence, for a spline of the form ($x$-$t$) and for the $SVMB_i$ variable, only a positive coefficient of the basis function is justified. However, taking into account the simultaneous impact of both basis functions on the dependent variable, it should be emphasized that the BF3 function has a smaller knot point value than BF7 (0.14% vs 0.347%) as well as a greater influence as measured by the absolute value of the regression coefficient, $|\beta_{BF3}| = 2.747 > |\beta_{BF7}| = 2.392$. This means that, considering the non-linear complexity of the MARS regression model, the final effect of

the $SVMB_i$ variable on the dependent variable is positive, especially when the value of the share in the VM balance is within the range (0.14%÷0.347).

The interpretation of the pair of basis functions BF3 and BF4 is the same as for the pair of basis functions BF1 and BF2.

The last pair of piece-wise regression spline functions is BF5 and BF6, which reveal the non-linear effect of the company's market capitalization delayed by one year ($MV_{(t-1)}$) on stock's return. A negative regression coefficient for the BF5 function indicates that if the market value at the beginning of the VM measurement process is greater than 3.684E+10 USD (the original value of normalized knot point is 0.04948) then the annualized rate of return on shares is lower, while the $MV_{(t-1)}$ is greater. On the other hand, if the $MV_{(t-1)}$ is below 3.684E +10 USD, then the rate of stock return increases while the $MV_{(t-1)}$ decreases, as implied by the positive correlation coefficient of BF6 spline function and the negative sign for $MV_{(t-1)}$ in the equitation. Construction of basis functions BF5 and BF6 and the values of their coefficients indicate the inverse relationships between the $MV_{(t-1)}$ and stock's returns, which can be summarized as follows:

for BF5:

$$\text{for } MV_{(t-1)} > 3.684E + 10 \text{ and } \nearrow MV_{(t-1)} \longrightarrow \nearrow BF5 \longrightarrow \searrow ARRS_i$$

$$\text{for } MV_{(t-1)} > 3.684E + 10 \text{ and } \searrow MV_{(t-1)} \longrightarrow \searrow BF1 \longrightarrow \nearrow ARRS_i$$

for BF6:

$$\text{for } MV_{(t-1)} < 3.684E + 10 \text{ and } \nearrow MV_{(t-1)} \longrightarrow \searrow BF5 \longrightarrow \searrow ARRS_i$$

$$\text{for } MV_{(t-1)} < 3.684E + 10 \text{ and } \searrow MV_{(t-1)} \longrightarrow \nearrow BF1 \longrightarrow \nearrow ARRS_i$$

where $\nearrow$ and $\searrow$ denote the increase and decrease of the variable, respectively.

The final regression equation of model 1 is as follows:

$$ARRS_i = 0.135 + 0.238 \cdot \max(0, \ CC_i - 0.714) - 0.225 \cdot \max(0, \ 0.714 - CC_i) + 2.747 \cdot$$
$$\max(0, \ SVMB_i - 0.014) - 4.618 \cdot \max(0, \ 0.014 - SVMB_i) - 0.209 \cdot \max(0, \ MV_{(t-1)} - 0.049) \quad (41)$$
$$+1.228 \cdot \max(0, \ 0.049 - MV_{(t-1)}) - 2.392 \cdot \max(0, \ SVMB_i - 0.035)$$

The results obtained from model 1 imply that the company's place on the self-organizing feature map in terms of its central position in the VMN and the share in VM balance have a positive effect on the dependent variable, and the market capitalization of company delayed by one year has a negative impact. The obtained results are presented in Fig 4.

The second model contains the same explanatory variables as model 1. The maximum number of basis functions and the penalty factor are also at the same level. Unlike model 1, a second-order interactions between predictors are allowed. Table 5 displays the parameters and MARS regression results for the second model.

From Table 5, one can find that, firstly, the overall GCV is lower compared to model 1 and equals 0.0048, and secondly, that model 2 explains the variability of the dependent variable to a greater extent (adjusted $R^2$ is 0.7147). The lower value of GCV and the explanation of over 71% of the variance of the annualized stock's return lead to the conclusion that model 2 is more robust compared to model 1.

The second model consists of 6 basis functions with a single spline and 4 basis functions with second-order interactions. Model 2 indicates that all independent variables play an important role in determining the annualized stock's return as all regression coefficients of

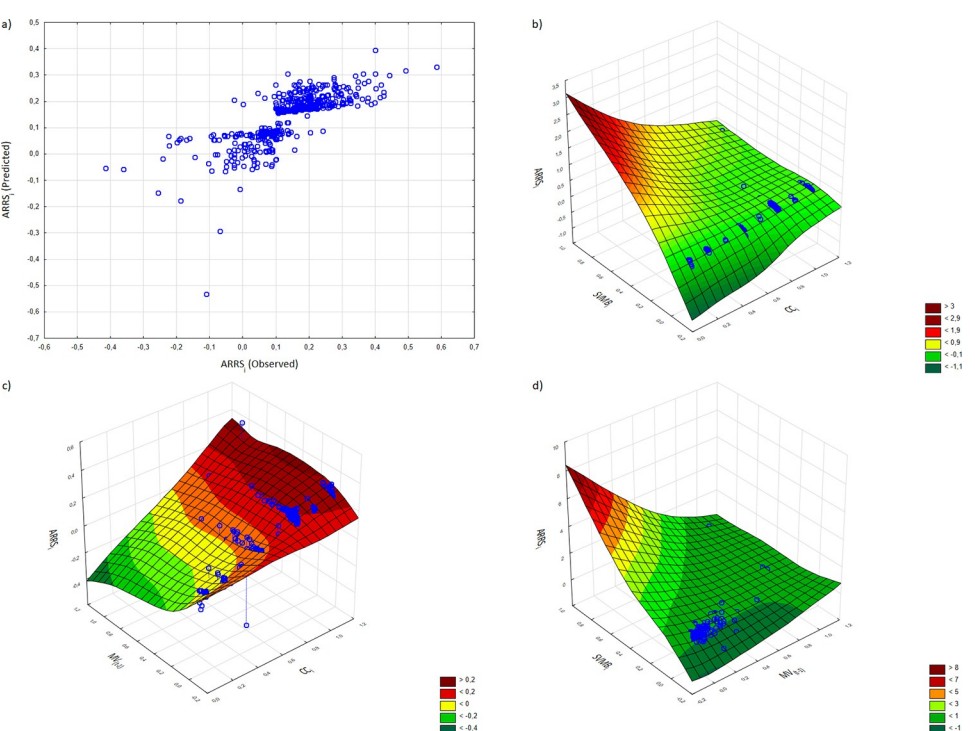

**Fig 4. Graphical illustration of MARS model 1.** (a) ARRS values observed versus predicted; (b) first-order term of the predictor variables $CC_i$ and $SVMB_i$; (c) first-order term of the predictor variables $CC_i$ and $MV_{(t-1)}$; (d) first-order term of the predictor variables $MV_{(t-1)}$ and $SVMB_i$ (color in print).

basis functions are statistically significant with the significance level set at 5%. The results are comparable to model 1.

The first four spline basis functions (BF1, BF2, BF3, BF4) have the same nonlinear form along with the values of the knot points as their counterparts in model 1. The only differences are the values of the regression coefficients of these functions, although the signs of the coefficients are preserved. This means that the independent variables–cluster of centrality and the share in VM balance–and the dependent variable move into the same direction, which confirms the conclusions from the analysis of model 1. It should be noted that the increase in the rate of return on shares applies to companies classified to clusters 6, 7, or 8, and their share in the value migration balance is greater than 0.14%.

The last pair of piece-wise spline functions in model 2 is BF9 and BF10, related with the variable $MV_{(t-1)}$. The construction of these functions is equivalent to the forms corresponding to the pair of basis functions in model 1 –BF5 and BF6, respectively. The function of the form ($x$-$t$) is associated with a negative regression coefficient and ($t$-$x$) relates to a positive regression coefficient. The difference concerns the values of these coefficients as well as the value of the knot point. The threshold is 1.948E+11 USD (the original value of BF9 and BF10 is 0.2617) and is much higher than $t = 3.685E+10$ USD for model 1. The negative relationship between $MV_{(t-1)}$ and stock's return applies to companies whose market capitalization at the beginning of the MV process measurement is greater than 1.948E+11 USD. Companies with relatively high market values find it more difficult to generate abnormal returns on stocks. In this context, the higher knot point values of the basis functions BF9 and BF10 in model 2, compared to their model 1 counterparts, are more suitable, increasing the reliability of model 2.

Considering the basis functions with second-order interactions (BF5÷BF8), three types of interactions were found: (i) interaction between $MV_{(t-1)}$ and $CC_i$ (BF5); (ii) interactions

**Table 5. Parameters and results for model 2.**

| MODEL 2 | | | | |
|---|---|---|---|---|
| **Parameters** | | Value | | |
| Dependent variable | | $ARRS_i$ | | |
| Independent variables | | $CC_i$; $SVMB_i$; $MV_{(t-1)}$ | | |
| Maximum number of basis functions | | 14 | | |
| Order of interaction | | 2 | | |
| Penalty | | 2 | | |
| Threshold | | 0.0005 | | |
| Removal of irrelevant basis functions | | Yes | | |
| Number of observations | | 471 | | |
| **Basis function** | | $\beta$ | *t* statistics | *p* |
| | Constant | 0.063 (0.018) | 3.393*** | 0.0008 |
| **BF1** | $\max(0,\ CC_i - 0.714)$ | 0.226 (0.063) | 3.595*** | 0.0004 |
| **BF2** | $\max(0,\ 0.714 - CC_i)$ | -0.164 (0.028) | -5.753*** | 0.0000 |
| **BF3** | $\max(0,\ SVMB_i - 0.014)$ | 1.011 (0.235) | 4.304*** | 0.0000 |
| **BF4** | $\max(0,\ 0.014 - SVMB_i)$ | -11.496 (0.679) | -16.931*** | 0.0000 |
| **BF5** | $\max(0,\ 0.072 - MV_{(t-1)}) \cdot \max(0,\ CC_i - 0.714)$ | 13.322 (1.342) | 9.925*** | 0.0000 |
| **BF6** | $\max(0,\ SVMB_i - 0.064) \cdot \max(0,\ CC_i - 0.714)$ | -1.365 (0.531) | -2.569* | 0.0105 |
| **BF7** | $\max(0,\ 0.064 - SVMB_i) \cdot \max(0,\ CC_i - 0.714)$ | -12.777 (1.783) | -7.165*** | 0.0000 |
| **BF8** | $\max(0,\ 0.014 - SVMB_i) \cdot \max(0,\ MV_{(t-1)} - 0.013)$ | 45.168 (3.898) | 11.588*** | 0.0000 |
| **BF9** | $\max(0,\ MV_{(t-1)} - 0.262)$ | -0.225 (0.075) | -2.988** | 0.0030 |
| **BF10** | $\max(0,\ 0.262 - MV_{(t-1)})$ | 0.654 (0.083) | 7.844*** | 0.0000 |
| **Results** | | Value | | |
| GCV | | 0.0048 | | |
| $R^2$ | | 0.7214 | | |
| Adjusted $R^2$ | | 0.7147 | | |

standard error in parentheses

***, ** and * denote significant levels at 0.1%, 1%, and 5%, respectively

between $SVMB_i$ and $CC_i$ (BF6, BF7); (iii) interaction between $SVMB_i$ and $MV_{(t-1)}$ (BF8). Interpretation of the results based on basis functions regarding second- or higher-order interactions is quite complex. However, the inclusion of the spline basis functions BF5÷BF8 in model 2 improves the explanation of the variance of the dependent variable.

The overall regression equation for model 2 is as follows:

$$
\begin{aligned}
ARRS_i &= 0.063 + 0.226 \cdot \max(0,\ CC_i - 0.714) - 0.164 \cdot \max(0,\ 0.714 - CC_i) + 1.011 \cdot \\
&\quad \max(0,\ SVMB_i - 0.014) - 11.496 \cdot \max(0,\ 0.014 - SVMB_i) + 13.322 \cdot \max(0,\ 0.072 - \\
&\quad MV_{(t-1)}) \cdot \max(0,\ CC_i - 0.714) - 1.365 \cdot \max(0,\ SVMB_i - 0.064) \cdot \max(0,\ CC_i - 0.714) - \\
&\quad 12.777 \cdot \max(0,\ 0.064 - SVMB_i) \cdot \max(0,\ CC_i - 0.714) + 45.168 \cdot \max(0,\ 0.014 - SVMB_i) \\
&\quad \cdot \max(0,\ MV_{(t-1)} - 0.013) - 0.225 \cdot \max(0,\ MV_{(t-1)} - 0.262) + 0.654 \cdot \max(0,\ 0.262 - \\
&\quad MV_{(t-1)})
\end{aligned} \tag{42}
$$

Graphical presentation of MARS model 2 is shown in Fig 5.

Table 6 presents the analysis of the importance of variables for models 1 and 2.

In the first model, the most important variable for the prediction of the annualized rate of return on shares is the cluster of centrality ($CC_i$). This implies that the place of the enterprise

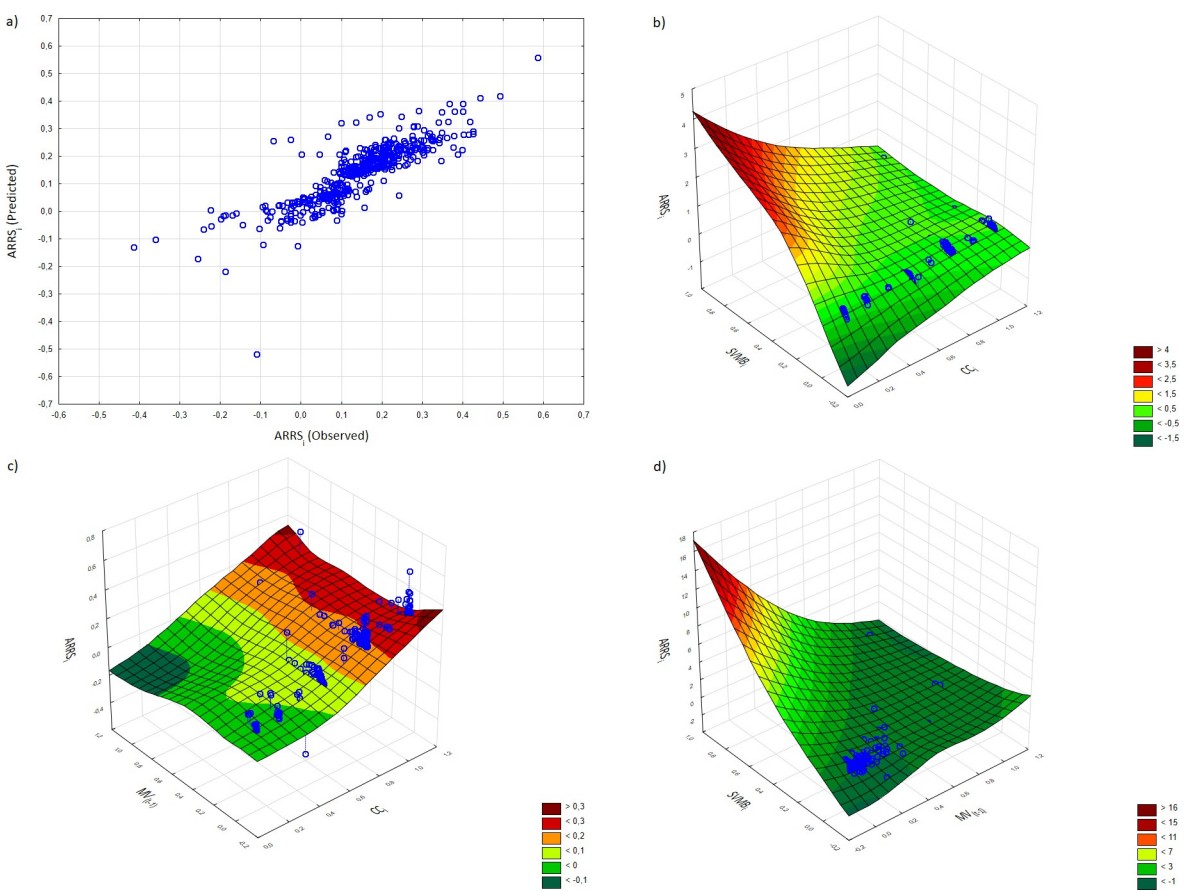

**Fig 5. Graphical illustration of MARS model 2.** (a) Observed versus predicted ARRS values; (b) second-order term of the predictor variables $CC_i$ and $SVMB_i$; (c) second-order term of the predictor variables $CC_i$ and $MV_{(t-1)}$; (d) second-order term of the predictor variables $MV_{(t-1)}$ and $SVMB_i$ (color in print).

on the self-organizing feature map of the central position in the VMN is the most dominant factor. Excluding the $CC_i$ variable from model 1 causes the $R^2$ statistic to drop from 0.5927 to 0.4154, and the $GCV$ to increase from 0.0068 to 0.0096. When the second-order interaction terms are considered in model 2, the most important variable for the prediction of the output variable is the share in value migration balance ($SVMB_i$). The importance of the $CC_i$ variable ranks third in the hierarchical order. Ultimately, the relative importance of $CC_i$ is over 53% of

**Table 6. Evaluation of the importance of the variables.**

| Variable | Basis functions | GCV | $R^2$ | Relative importance |
|---|---|---|---|---|
| **Model 1** | | | | |
| $CC_i$ | BF1, BF2 | 0.0096 | 0.4154 | 100.00% |
| $SVMB_i$ | BF3, BF4, BF7 | 0.0085 | 0.4685 | 62.80% |
| $MV_{(t-1)}$ | BF5, BF6 | 0.0071 | 0.5612 | 10.65% |
| **Model 2** | | | | |
| $CC_i$ | BF1, BF2, BF5, BF6, BF7 | 0.0067 | 0.5946 | 53.92% |
| $SVMB_i$ | BF3, BF4, BF6, BF7, BF8 | 0.0085 | 0.4685 | 100.00% |
| $MV_{(t-1)}$ | BF5, BF8, BF9, BF10 | 0.0071 | 0.5651 | 64.99% |

**Table 7. Linear regression analysis results.**

| Variable | Model 3 | Model 4 |
|---|---|---|
| Constant | -0.092 *** (-0.092) | -0.102 *** (0.015) |
| $CC_i$ | 0.315 *** (0.017) | 0.314 *** (0.019) |
| $SVMB_i$ | 0.769 *** (0.124) | 3.319 *** (0.629) |
| $MV_{(t-1)}$ | -0.309 *** (0.057) | -0.054 (0.177) |
| $MV_{(t-1)}{}^*CC_i$ | | -0.403 (0.215) |
| $SVMB_i{}^*CC_i$ | | -1.648 * (0.751) |
| $SVMB_i{}^*MV_{(t-1)}$ | | -0.643 (0.331) |
| Number of observations ($n$) | 471 | 471 |
| $F$-test | 161.778 *** | 99.326 *** |
| $R^2$ | 0.5096 | 0.5622 |
| Adjusted $R^2$ | 0.5065 | 0.5566 |
| Mean VIF | 2.158 | 48.647 |
| Max. VIF | 2.760 | 127.798 |

standard error in parentheses

***, ** and * denote significant levels at 0.1%, 1%, and 5%, respectively

the most dominant variable, which means that taking into account the more complex model with nonlinear form of the spline functions with interaction effects, all three dependent variables are significant in the modeled relationship. When $CC_i$ is removed from model 2, the $R^2$ statistic is reduced from 0.7214 to 0.5946, while the $GCV$ value increases from 0.0048 to 0.0067. It can therefore be concluded that the classification of the company into a specific group due to its central position in the VMN network is a significant indicator of the stock's return.

Finally, it should be noted that the obtained results of both models are the best for the penalty factor equal to 2 (most often in MARS approach the penalty factor is set to 2 or 3).

OLS regression analysis was performed for comparison. The results of the linear regression estimation are shown in Table 7.

Model 3 contains the same variables as Model 1, with no interactions between the variables. All variables are statistically significant at the level $p = 0.001$, and the regression coefficient for $CC_i$ and $SVMB_i$ is positive, while for $MV_{(t-1)}$ it is negative. The linear model (Model 3) explains the variability of the dependent variable to a lesser extent compared to the MARS model (Model 1), adjusted $R^2$ are 0.5065 and 0.5856, respectively. However, the linear model confirms a statistically significant influence of the cluster of centrality on stock's return.

For a linear model, the importance of the predictors can be assessed by maximizing the absolute value of the standardized regression coefficient $|\beta|$. The order of importance of the variables is as follows (in parentheses $|\beta|$): 1) $CC_i$ (0.624); 2) $SVMB_i$ (0.334); $MV_{(t-1)}$ (0.284). This confirms the conclusion of the significance of the cluster of centrality, drawn from the MARS model analysis.

Model 4 is a linear representation of the MARS model 2, where the individual independent variables and their interactions are included–(i) $MV_{(t-1)}{}^*CC_i$; (ii) $SVMB_i{}^*CC_i$; (iii) $SVMB_i{}^*MV_{(t-1)}$. However, only 3 variables are statistically significant, including the $CC_i$ variable ($p<0.001$), for which the regression coefficient is positive. The linear model 4 offers less ability to explain the variability than the corresponding MARS model 2, with the adjusted $R^2$ at 0.5566 and 0.7214, respectively. Hence, MARS regression models are more robust than linear models. Moreover, model 4 displays the collinearity problem of independent variables, which is indicated by a considerably exceeded acceptable level of the mean and maximum value of variance inflation factor (VIF>10).

## 5. Additional analysis

The limitation of the VMN built for 498 stocks is the imperfect projection of the full complexity for such a number of companies in the adopted one-year period. The value migration process should be analyzed over a properly defined period. It should not be too short in order to eliminate short-term fluctuations in stock prices. However, it should not be too long either to capture the scale of the process of changing the value of companies. The adopted period of one year results from the need to capture the current value migration process. In this section, the robustness of obtained results is carried out to narrow the research sample to the 250 largest companies from the S&P500 index. A similar approach was used in the study [110] to investigate the statistical properties of price returns for the 100 largest companies on the NYSE in one year, 2002. The aim of the analysis is to assess whether the above limitation biases the obtained results.

The largest companies are considered in terms of market capitalization in 2019. The analysis period remains the same; therefore, the dataset consists of 253 daily closing prices in every 250 time series. The value migration network for the 250 largest companies is presented in Fig A1, available in S1 File. The largest component of the VMN contains 219 firms.

The obtained topological layer of the SOM with the assumed dimensions of 3 x 4 again contains eight disjoint groups. The number of companies assigned to individual clusters is presented in Table A1 and Fig A2a in S1 File (see S1 File). The calculation of the cluster of centrality ($CC_i$) variable is provided in Table A2 in S1 File (see S1 File.).

The results of the MARS regression without the interaction of variables (model 5) are presented in Table 8. The final regression equation of model 5 is as follows:

$$ARRS_i = 0.221 + 0.085 \cdot \max(0, \ CC_i - 0.714) - 0.264 \cdot \max(0, \ 0.714 - CC_i) + 1.747 \cdot$$
$$\max(0, \ SVMB_i - 0.033) - 3.504 \cdot \max(0, \ 0.033 - SVMB_i) - 0.227 \cdot \max(0, \ MV_{(t-1)} - 0.076) \ (43)$$
$$+ 1.925 \cdot \max(0, \ 0.076 - MV_{(t-1)}) - 1.267 \cdot \max(0, \ SVMB_i - 0.064)$$

The graphical presentation of model 5 is demonstrated in Fig A3 in S1 File.

Model 5 is the counterpart of Model 1. By comparing both models, it can be seen that the structure of all the spline basis functions and the signs (positive/negative) of the regression coefficients are preserved. Furthermore, all regression coefficients of the contained spline basis functions are statistically significant ($p < 0.05$). The value of the knot point of BF1 and BF2 is the same as in the original model: the value of 0.714 normalized knot; the value of 5 $CC_i$ variable. This means that the critical threshold of the cluster for the non-linear effect remains unchanged. However, due to the change in the research sample, the knot point values have changed for the remaining functions related to the $SVMB_i$ and $MV_{(t-1)}$ variables.

The results of Model 6 with second-order interactions between predictors are shown in Table 9.

Model 6 is the counterpart of Model 2. It consists of fewer base functions, six functions related to three variables, and one function with a second-order interaction. The following basic functions of model 6 (BF1÷BF7) are equivalent to the basic functions of BF1, BF2, BF3, BF4, BF9, BF10, and BF8 of model 2, respectively. The construction of all basis functions and the signs of the regression coefficients are preserved and statistically significant at the significance level of 5%. In this case, the value of the knot point of BF1 and BF2 is the same as in the original model 2 and for models 1 and 5.

**Table 8. Parameters and results for model 5.**

| MODEL 5 | | | |
|---|---|---|---|
| **Parameters** | | **Value** | |
| Dependent variable | | $ARRS_i$ | |
| Independent variables | | $CC_i$; $SVMB_i$; $MV_{(t-1)}$ | |
| Maximum number of basis functions | | 14 | |
| Order of interaction | | 1 | |
| Penalty | | 2 | |
| Threshold | | 0.0005 | |
| Removal of irrelevant basis functions | | Yes | |
| Number of observations | | 219 | |
| **Basis function** | | **$\beta$** | **$t$ statistics** | **$p$** |
| | Constant | 0.221 (0.013) | 16.854*** | 0.0000 |
| BF1 | $\max(0, CC_i - 0.714)$ | 0.085 (0.063) | 1.343* | 0.0487 |
| BF2 | $\max(0, 0.714 - CC_i)$ | -0.264 (0.042) | -5.834*** | 0.0000 |
| BF3 | $\max(0, SVMB_i - 0.033)$ | 1.747 (0.666) | 2.622** | 0.0094 |
| BF4 | $\max(0, 0.033 - SVMB_i)$ | -3.504 (0.408) | -8.592*** | 0.0000 |
| BF5 | $\max(0, MV_{(t-1)} - 0.076)$ | -0.227 (0.061) | -3.696*** | 0.0003 |
| BF6 | $\max(0, 0.076 - MV_{(t-1)})$ | 1.925 (0.298) | 6.466*** | 0.0000 |
| BF7 | $\max(0, SVMB_i - 0.064)$ | -1.267 (0.675) | -1.877* | 0.0491 |
| **Results** | | **Value** | | |
| GCV | | 0.0061 | | |
| $R^2$ | | 0.6755 | | |
| Adjusted $R^2$ | | 0.6631 | | |

standard error in parentheses

***, ** and * denote significant levels at 0.1%, 1%, and 5%, respectively

The final regression equation of model 5 is as follows:

$$ARRS_i = 0.240 + 0.085 \cdot \max(0,\ CC_i - 0.714) - 0.169 \cdot \max(0,\ 0.714 - CC_i) + 0.710 \cdot$$
$$\max(0,\ SVMB_i - 0.033) - 6.612 \cdot \max(0,\ 0.033 - SVMB_i) - 0.331 \cdot \max(0,\ MV_{(t-1)} -$$
$$0.076) + 2.776 \cdot \max(0,\ 0.076 - MV_{(t-1)}) + 19.963 \cdot \max(0,\ MV_{(t-1)} - 0.027) \cdot$$
$$\max(0,\ 0.033 - SVMB_i) \tag{44}$$

Theraphic presentation of MARS model 6 is shown in Fig A4 in S1 File.

In summary, the robustness checks carried out demonstrated that the main findings are robust to the revised research sample in the form of a limitation to the 250 largest companies in the S&P500 index. Value migration can be analyzed in relation to a specified system of enterprises, e.g., the entire market, its selected part, or industry. The results obtained for the group of 250 largest companies are consistent with those for all companies from the S&P500 index. However, this does not mean that the value migration occurs separately within one of the two groups, the largest or the smallest companies. In other words, it is possible that the value flows between a company assigned to the TOP 250 group and smaller ones. However, the VMN built on the basis of all 498 stocks covers a wider range of value migration in the financial market.

## 6. Conclusions

In this work, a complex network has been formed using data for 498 companies from S&P500 index from 31 December 2018 to 31 December 2019. The value migration network

**Table 9. Parameters and results for model 6.**

| MODEL 6 | | | | |
|---|---|---|---|---|
| **Parameters** | | Value | | |
| Dependent variable | | $ARRS_i$ | | |
| Independent variables | | $CC_i$; $SVMB_i$; $MV_{(t-1)}$ | | |
| Maximum number of basis functions | | 14 | | |
| Order of interaction | | 2 | | |
| Penalty | | 2 | | |
| Threshold | | 0.0005 | | |
| Removal of irrelevant basis functions | | Yes | | |
| Number of observations | | 219 | | |
| **Basis function** | | $\beta$ | *t* statistics | *p* |
| | Constant | 0.240 (0.015) | 14.511*** | 0.0000 |
| **BF1** | max(0, $CC_i$– 0.714) | 0.085 (0.064) | 1.474* | 0.0492 |
| **BF2** | max(0, 0.714 –$CC_i$) | -0.169 (0.042) | -6.105*** | 0.0000 |
| **BF3** | max(0, $SVMB_i$– 0.033) | 0.710 (0.152) | 4.528*** | 0.0000 |
| **BF4** | max(0, 0.033 –$SVMB_i$) | -6.612 (0.384) | -9.887*** | 0.0000 |
| **BF5** | max(0, $MV_{(t-1)}$– 0.076) | -0.331 (0.083) | -3.732*** | 0.0002 |
| **BF6** | max(0, 0.076 –$MV_{(t-1)}$) | 2.776 (0.363) | 6.048*** | 0.0000 |
| **BF7** | max(0, 0.033 –$SVMB_i$) · max(0, $MV_{(t-1)}$– 0.027) | 19.963 (0.379) | 1.858* | 0.0376 |
| **Results** | | Value | | |
| GCV | | 0.0051 | | |
| $R^2$ | | 0.7273 | | |
| Adjusted $R^2$ | | 0.7170 | | |

standard error in parentheses

***, ** and * denote significant levels at 0.1%, 1%, and 5%, respectively

constructed with the use of MST-Pathfinder has been employed as a tool to analyze the relationship between VM process and stock returns. Specifically, a complex network, artificial neutral network, and MARS regression have been applied to investigate the relationship between centrality measures of the VMN and individual stock returns. This paper is the first study that incorporates the above three approaches in the analysis of real complex financial systems.

This study explored the relationship between self-organizing feature map representing the centrality of a company's position in the VMN, and stock's return. Specifically, the investigation was focused on whether the central position of the company in the VMN affects the annualized rate of return on shares. The results obtained from the empirical analysis show that the topological position in the VMN has a statistically significant effect on the stock's returns. This study appears to be the first to demonstrate that value migration, measured by means of a self-organizing map based on the centrality measures of the VMN, plays an important role in determining stock's return levels.

Both MARS regression models show a non-linear effect of the company's centrality position in the VMN operationalized by the SOM on the annualized rate of return on shares. For firms classified into neurons (3,2); (3,1), and (2,1)–with CCi at 6, 7, or 8 –the company's place on the self-organizing feature map in terms of its central position in the VMN has a positive effect on stock's return and for companies from neurons (2,4); (1;4); (3,4); (1,3)–with CCi at 1, 2, 3, or 4 –a negative effect is demonstrated. However, the absolute values of the regression coefficients of the basis functions are different, which means the diverse strength of the influence of the

variable centrality of cluster ($CC_i$) on the stock's return. In conclusion, the results of this study reveal that the level of centrality in the VMN is a non-linear indicator of the annualized rate of return on shares. This is confirmed by worse results in explaining the variability of the dependent variable by comparable linear models.

This study contributes to the area of (i) complex network; (ii) portfolio optimization, (iii) value-based management. Firstly, these results emphasize the importance of financial filtered networks and the position of companies in the value migration network. The place of the enterprise on the SOM is not only a statistically significant factor affecting the annualized rate of return on shares, but also shows high relative importance in the MARS regression models. For model 1, the classification of the firm into a cluster of network centrality is the most important determinant of the dependent variable. For model 2, the relative importance of clusters according to network centrality is over 53% of the most dominant variable.

Secondly, in the context of building an optimal stock portfolio strategy, it cannot be clearly indicated whether it is better to invest in central or peripheral stocks. The resulting classification using the Kohonen network and the four centrality measures does not determine whether the analyzed assets are central or peripheral. However, previous study [59] has revealed the diversifying impact of the centrality level in VMN on the rate of return on shares. This is due to the different nature of the individual centrality measures used, for which a positive effect was demonstrated for in-degree centrality, and a negative one for out-degree and entropy centrality. In other words, depending on the different type of centrality measures, better results in terms of the rate of return on shares are achieved by central or peripheral companies. It should be pointed out that the obtained results cannot be directly compared with the results in terms of the level of stock's return achieved by a set of stocks or a portfolio consisting of central or peripheral assets. Previous studies have been conducted with the use of different centrality measures dedicated to undirected networks (degree, closeness, betweenness, eigenvector centrality), which is the cross-correlation of log-return of stock price network. However, SOM based on VMN centrality can be useful in constructing the optimal portfolio.

Thirdly, one potential application of this work is to support value-based management evaluation tools. The topological position in the value migration network has an opportunity to become a new assessment measure in terms of the efficiency of implementing the concept of value-based management. Companies capable of intercepting the value in the financial market will be ranked higher on the SOM of network centrality. Consequently, enterprises in the inflow phase achieve higher rates of return on shares, providing a reliable view on the achievement of the company's goal of maximizing the value of the enterprise.

Future studies should focus on the impact of the type of sector on the VM process. Value may flow between separate companies, but the process of value fluctuation can be considered at a higher level of aggregation, i.e., economic sectors. Further research is needed to investigate value migration in a dynamic approach, where the subject of the study is a dynamic network. Such a network pattern will enable the analysis of the intensity of changes in the value migration process. However, parameters such as the window length and the rolling step must be properly set to use the rolling window approach for the value migration network.

## Supporting information

**S1 File. Supporting information for: The effect of self-organizing map architecture based on the value migration network centrality measures on stock return.** Evidence from the US market. The Supporting Information, S1 File, contains additional tables and graphs for the robustness analysis.
(PDF)

**S1 Dataset.**
(XLSX)

## Author Contributions

**Conceptualization:** Dariusz Siudak.

**Data curation:** Dariusz Siudak.

**Formal analysis:** Dariusz Siudak.

**Investigation:** Dariusz Siudak.

**Methodology:** Dariusz Siudak.

**Validation:** Dariusz Siudak.

**Visualization:** Dariusz Siudak.

**Writing – original draft:** Dariusz Siudak.

**Writing – review & editing:** Dariusz Siudak.

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
