## [Decision Letter · Decision Letter 0]

31 Aug 2022

PONE-D-22-15643The effect of self-organizing map architecture based on the value migration network centrality measures on stock return. Evidence from the US marketPLOS ONE

Dear Dr. Siudak,

Thank you for submitting your manuscript to PLOS ONE. After careful consideration, we feel that it has merit but does not fully meet PLOS ONE’s publication criteria as it currently stands. Therefore, we invite you to submit a revised version of the manuscript that addresses the points raised during the review process.

Sorry for the delay in the review process; too many potential reviewers have been busy in the summer period. The recommendations provided by two reviewers probably might help improve the paper.

We look forward to receiving your revised manuscript.

Kind regards,

Vygintas Gontis, Ph.D.

Academic Editor

PLOS ONE

Journal Requirements:

2. Please upload a new copy of Figure 1, 2, 3, 4 and 5 as the detail is not clear. Please follow the link for more information: https://blogs.plos.org/plos/2019/06/looking-good-tips-for-creating-your-plos-figures-graphics/

3. Please upload a copy of Supporting Information S1 Fig 1 to S5 Fig 5  which you refer to in your text on page 42 and 43.

Reviewers' comments:

Reviewer's Responses to Questions

**Comments to the Author**

1. Is the manuscript technically sound, and do the data support the conclusions?

Reviewer #1: Partly

Reviewer #2: Partly

2. Has the statistical analysis been performed appropriately and rigorously? 

Reviewer #1: No

Reviewer #2: No

3. Have the authors made all data underlying the findings in their manuscript fully available?

Reviewer #1: Yes

Reviewer #2: Yes

4. Is the manuscript presented in an intelligible fashion and written in standard English?

Reviewer #1: Yes

Reviewer #2: Yes

5. Review Comments to the Author

Reviewer #1: In this manuscript the network measures are used for studying the value migration process based on the stock data of 498 companies listed in the S&P500 index between the end of 2018 to the end of 2019 which results in 253 daily closing prices in each time-series. The first related question is why just this period? Even more importantly, however, using such a collection of 498 time series of length 253 results in a 498 X 498 correlation matrix (introduced in section 2.1) whose real rank is 253 thus produces 498-253 = 245 zero modes. Expressing this in simple terms such a collection of data points is too poor to project out the full complexity of correlations among the 498 companies. This makes the results presented unreliable. This immediately detectable drawback needs to properly corrected/elaborated before we can proceed any further with the review procedure.

Also, about a dacade ago exactly the same market within the same methodology (of MST) has been studies in 'Physical approach to complex systems' doi:10.1016/j.physrep.2012.01.007 (Figs 80-81), and the central hub was the General Electric. It would be interesting to compare and explicitely address this point here. By the way, the corresponding figure 2 of the present manuscript is hardly readable. What makes commenting also difficult is that the equations are not numbered here.

Reviewer #2: This is an interesting manuscript using variety of techniques (including trendy machine learning algorithms) to explore the impact of the stock centrality in the value migration network onto its annual return. While combination of methods used in this paper is rather broad and interesting, I still have some doubts about novelty and non-triviality of the reported results.

My main issue is that when building VMN information about market returns seems to be used. I am talking about step 4 (lines 197-198) of the VMN formation algorithm. Doesn't this make results trivial? Annual returns (via market capitalization) inform VMN and then VMN informs annual returns?

The introduction of the manuscript starts with discussion on VM process (i.e., something dynamic). Yet later in the paper VMN is treated as static. Can't the stocks position in the VMN change over time?

Initial paragraphs of Section 2 are written in confusing manner. It is not fully clear what is used as input for VMN construction and what is the output. Similarly it is not clear what is the input for Kohonen network and how should we interpret its output. More emphasis on "input"->"process"->"output" should be made, so that it would be clear for readers unfamiliar with specificities of each process.

Is daily return data sufficient for this task? What would change if the data would be of higher/lower frequency?

Most figures are unreadable. Resolution is extremely low - can't make out what is on the axes.

It seems that term "multidimensional centrality" is used somewhat confusingly. Do you mean "network centrality" in the VMN? Which of the centrality measures do you have in mind, or the results do not depend on the centrality measure? Typically "multidimensional centrality" is centrality of a node in multidimensional (multilayer) network, but here we have no such networks.

Lines 69-70: VMN hasn't yet been introduced, but we are already discussing its properties. At this point it is hard to comprehend what does it imply for the VMN network to have power-law degree distribution.

Lines 77-78: Requires citation?

Lines 157-159: Sentence makes no sense. How can measures perform? What do they perform?

Lines 178-188: MTS -> MST. This change likely will need to be made in other parts of the manuscript, too.

Line 202: Why? Wouldn't it be more reasonable to use unit threshold value?

Line 207: Is it possible that both i and j are in inflow phase? How can this happen? What does it mean?

Line 436: Figure seems to contain undirected network, yet the text seems to imply that VMN is directed network. Or is it poor quality of the figure?

Table 1: What do you mean by "error level". Your data seems to be unlabeled and you are using unsupervised machine learning algorithm. How do you define error?

Line 681-684: What does it mean for a firm to have a positive impact on stock's return? Confusing sentence.

6. PLOS authors have the option to publish the peer review history of their article (what does this mean?). If published, this will include your full peer review and any attached files.

Reviewer #1: No

Reviewer #2: No

---

## [Author Response · Author response to Decision Letter 0]

26 Sep 2022

Response to Academic Editor:

Thank you very much for the referee report on my manuscript PONE-D-22-15643 entitled “The effect of self-organizing map architecture based on the value migration network centrality measures on stock return. Evidence from the US market”. I have revised the paper to fully address all Reviewer comments.

Please find enclosed a detailed, point-to-point response to all Reviewer comments with all changes clearly specified. Changes in the text are marked with colors, as follows: Reviewer #1: green; Reviewer #2: yellow. Minor inaccuracies have been found during editing and correction. They have been rectified and marked in blue. All page references are related to the manuscript with the marked changes.

I wish to take this opportunity to thank the Reviewers for their time, effort and insightful comments that have resulted in an improved manuscript. I would also like to thank you for handling my paper. Hopefully, my revised manuscript can be regarded as meeting the high standards of PLOS ONE journal.

With kind regards

Response to Reviewer #1:

I am grateful for the comments provided by the Reviewer to improve my manuscript. In my new version, I have followed the recommendations and your comments have been fully addressed. Text changes are marked in green. The newly added and significantly revised portions in the paper are also marked in green to make them easy to identify. Minor inaccuracies have been found during editing and correction. They have been rectified and marked in blue. All page references are related to the manuscript with the marked changes.

Response to Reviewer #2:

I am very grateful to the reviewer for his/her valuable comments, which have guided me in revising the paper. Text changes are marked in yellow. The newly added and significantly revised portions in the paper are also marked in yellow to make them easy to identify. Minor inaccuracies have been found during editing and correction. They have been rectified and marked in blue. All page references are related to the manuscript with the marked changes.

---

## [Decision Letter · Decision Letter 1]

2 Oct 2022

PONE-D-22-15643R1The effect of self-organizing map architecture based on the value migration network centrality measures on stock return. Evidence from the US marketPLOS ONE

Dear Dr. Siudak,

Thank you for submitting your manuscript to PLOS ONE. After careful consideration, we feel that it has merit but does not fully meet PLOS ONE’s publication criteria as it currently stands. Therefore, we invite you to submit a revised version of the manuscript that addresses the points raised during the review process. Please submit your revised manuscript by Nov 16 2022 11:59PM. If you will need more time than this to complete your revisions, please reply to this message or contact the journal office at plosone@plos.org. Please include the following items when submitting your revised manuscript:A rebuttal letter that responds to each point raised by the academic editor and reviewer(s). You should upload this letter as a separate file labeled 'Response to Reviewers'.A marked-up copy of your manuscript that highlights changes made to the original version. You should upload this as a separate file labeled 'Revised Manuscript with Track Changes'.An unmarked version of your revised paper without tracked changes. You should upload this as a separate file labeled 'Manuscript'.

We look forward to receiving your revised manuscript.

Kind regards,

Vygintas Gontis, Ph.D.

Academic Editor

PLOS ONE

Additional Editor Comments:

Could you be more careful addressing the recommendations of reviewer 1? Reviewer requests have to be implemented in the new version of the manuscript. 

Reviewers' comments:

Reviewer's Responses to Questions

**Comments to the Author**

1. If the authors have adequately addressed your comments raised in a previous round of review and you feel that this manuscript is now acceptable for publication, you may indicate that here to bypass the “Comments to the Author” section, enter your conflict of interest statement in the “Confidential to Editor” section, and submit your "Accept" recommendation.

Reviewer #1: (No Response)

Reviewer #2: All comments have been addressed

2. Is the manuscript technically sound, and do the data support the conclusions?

Reviewer #1: Partly

Reviewer #2: Yes

3. Has the statistical analysis been performed appropriately and rigorously? 

Reviewer #1: N/A

Reviewer #2: Yes

4. Have the authors made all data underlying the findings in their manuscript fully available?

Reviewer #1: (No Response)

Reviewer #2: Yes

5. Is the manuscript presented in an intelligible fashion and written in standard English?

Reviewer #1: (No Response)

Reviewer #2: Yes

6. Review Comments to the Author

Reviewer #1: The Author's response concerning my quantitative arguments related to the degeneracy of so-constructed correlation matrix expressed in my previous report is not satisfactorily expressed. This issue is now addressed, but as far as I can see only in the response letter and not in the manuscript, and, in addition, it is too descriptive. This issue should be made more quantitative and addressed in the manuscript text. In particular, does the claim that reducing the number of S&P500 companies used to the 250 largest ones does not change the result shown does mean that the remaing ones are disonnected from those 250 largest ones? If so, what for to include them?

Reviewer #2: (No Response)

7. PLOS authors have the option to publish the peer review history of their article (what does this mean?). If published, this will include your full peer review and any attached files.

Reviewer #1: No

Reviewer #2: No

---

## [Author Response · Author response to Decision Letter 1]

4 Oct 2022

Response to Academic Editor:

The reviewer's recommendations have been implemented in the new version of the manuscript.

Response to Reviewer #1:

I am grateful for your suggestions to improve my manuscript. In my new version, I have followed your recommendations. Changes in the text are marked with colors, as follows: 

first round of review: green; 

second round of review: grey.

I hope my revised manuscript meets your high standards.

One of the limitations of the study is the imperfect projection of the full complexity, which was highlighted in the previous review. The goal of the additional analysis was to assess whether the indicated limitation causes a distortion of the basic results. These analyzes are presented in Section 5 and in the Appendix (S1 File. Supporting information). 

The reduction of the analyzed companies to TOP250 in terms of market capitalization did not change the results in relation to all 498 companies. This does not mean, however, that value migration takes place severally and independently of each other in the group of only the largest and smallest stocks. Value migration can be analyzed in relation to a specified system of companies, limited do the entire market or its selected part (e.g., TOP100 or TOP250). However, the inclusion of all 498 companies was intended to capture the wide-ranging process of value migration in the financial market.

Section 5 has been extended at the beginning and end of this section.

---

## [Decision Letter · Decision Letter 2]

10 Oct 2022

The effect of self-organizing map architecture based on the value migration network centrality measures on stock return. Evidence from the US market

PONE-D-22-15643R2

Dear Dr. Siudak,

We’re pleased to inform you that your manuscript has been judged scientifically suitable for publication and will be formally accepted for publication once it meets all outstanding technical requirements.

Kind regards,

Vygintas Gontis, Ph.D.

Academic Editor

PLOS ONE

Additional Editor Comments (optional):

Reviewers' comments:

Reviewer's Responses to Questions

**Comments to the Author**

1. If the authors have adequately addressed your comments raised in a previous round of review and you feel that this manuscript is now acceptable for publication, you may indicate that here to bypass the “Comments to the Author” section, enter your conflict of interest statement in the “Confidential to Editor” section, and submit your "Accept" recommendation.

Reviewer #1: All comments have been addressed

2. Is the manuscript technically sound, and do the data support the conclusions?

Reviewer #1: Yes

3. Has the statistical analysis been performed appropriately and rigorously? 

Reviewer #1: Yes

4. Have the authors made all data underlying the findings in their manuscript fully available?

Reviewer #1: Yes

5. Is the manuscript presented in an intelligible fashion and written in standard English?

Reviewer #1: Yes

6. Review Comments to the Author

Reviewer #1: The concerns expressed before are now properly addressed and thus this manuscript can be accepted for publication in PLOS One its current form.

7. PLOS authors have the option to publish the peer review history of their article (what does this mean?). If published, this will include your full peer review and any attached files.

Reviewer #1: No

---

## [Editor Report · Acceptance letter]

18 Oct 2022

PONE-D-22-15643R2 

The effect of self-organizing map architecture based on the value migration network centrality measures on stock return. Evidence from the US market 

Dear Dr. Siudak:

I'm pleased to inform you that your manuscript has been deemed suitable for publication in PLOS ONE. Congratulations! Your manuscript is now with our production department. 

Kind regards, 

on behalf of

Dr. Vygintas Gontis 

Academic Editor

PLOS ONE